# Fungal Melanins and Applications in Healthcare, Bioremediation and Industry

**DOI:** 10.3390/jof7060488

**Published:** 2021-06-18

**Authors:** Ellie Rose Mattoon, Radames J. B. Cordero, Arturo Casadevall

**Affiliations:** 1Krieger School of Arts and Sciences, Johns Hopkins University, Baltimore, MD 21218, USA; emattoo1@jhu.edu; 2Department of Molecular Microbiology and Immunology, Johns Hopkins Bloomberg School of Public Health, 615 North Wolfe Street, Baltimore, MD 21205, USA; acasade1@jhu.edu

**Keywords:** industrial microbiology, melanin, fungi, radioprotection, biotechnology, fungal pigments

## Abstract

Melanin is a complex multifunctional pigment found in all kingdoms of life, including fungi. The complex chemical structure of fungal melanins, yet to be fully elucidated, lends them multiple unique functions ranging from radioprotection and antioxidant activity to heavy metal chelation and organic compound absorption. Given their many biological functions, fungal melanins present many possibilities as natural compounds that could be exploited for human use. This review summarizes the current discourse and attempts to apply fungal melanin to enhance human health, remove pollutants from ecosystems, and streamline industrial processes. While the potential applications of fungal melanins are often discussed in the scientific community, they are successfully executed less often. Some of the challenges in the applications of fungal melanin to technology include the knowledge gap about their detailed structure, difficulties in isolating melanotic fungi, challenges in extracting melanin from isolated species, and the pathogenicity concerns that accompany working with live melanotic fungi. With proper acknowledgment of these challenges, fungal melanin holds great potential for societal benefit in the coming years.

## 1. Introduction

The term melanin refers to a diverse set of dark polymeric pigments found in all kingdoms of life. In fungi, melanin plays a panoply of protective roles against stress (Table 1; Figure 1) [1]. For example, fungal melanin can protect against ionizing radiation, including ultraviolet, X-ray, gamma-ray, and particulate radiation [2,3,4]. In addition, melanin can play a role in thermoregulation and protection against both heat and cold shock [5,6]. For example, melanized *Cryptococcus neoformans* cells were more likely to survive both heat shock and cold shock compared to non-melanized species [6]. Melanized endophytes in a mutualistic relationship with plants will often help their symbionts thermoregulate by dissipating heat and absorbing ROS [7]. Darkly pigmented yeasts and mushrooms tend to be more common at higher absolute latitudes and colder climates, suggesting that the pigment’s ability to capture radiation energy and dissipate it as heat provides an advantage in generating thermal energy [5,8]. Part of melanin’s role in thermotolerance may be attributable to its ability to react with and neutralize Reactive Oxygen Species (ROS), helping fungal organisms withstand the oxidative stress that often accompanies higher temperatures [9].

In addition, melanin can help fungi withstand chemical stressors. In halotolerant black yeast, melanin synthesis inhibitors diminished the yeast’s ability to grow in hypersaline environments [10]. Researchers posited that this could be due to the stabilizing effect that melanin has on the fungal cell wall, which would have permitted a more effective response to osmotic stress [10]. In addition, melanization has also been shown to protect fungal cells from heavy metal stress and hydrolytic enzymes [11,12]. In areas with low water content, melanin is associated with stress response to dry conditions. For example, heavy melanization was associated with survival in microcolonial rock fungi exposed to long periods of desiccation [13]. Some fungal species in spalted woods will melanize in response to periods of low water content, creating black zones in the wood [14]. These numerous protective functions enable melanized fungi to reside in some of civilization’s most extreme environments, from deep-sea vents to the International Space Station [15,16].

Given the numerous functional groups present in melanotic pigments, melanin can bind and interact with many different organic and inorganic molecules [1,17]. One notable example is melanin’s affinity for chelating metal ions, which can be toxic to fungal cells [17,18,19]. Fungal melanin also plays a role in protecting the fungus against the human immune response and is often associated with pathogenicity in *Aspergillus fumigatus*, *Cryptococcus neoformans*, and *Talaromyces marneffei*, among others [20].

Lastly, melanin is correlated with fungal virulence and has been implicated as a possible antifungal target [21]. While fungal melanin’s role in virulence will be limited in this review, please see [20] for a more thorough overview of this topic.

Some of the functions that melanin holds in biology can be utilized for societal benefit. These biological functions of radioprotection, stress response, and substrate binding, among others, often stem from melanin’s unique physical and chemical properties. When properly understood, scientists may be able to use melanin to carry out parallel purposes in the fields of industry, healthcare, and bioremediation.

## 2. Fungal Melanins

Investigators wishing to work with melanin have several different options for sourcing the material, whether from animals such as cuttlefish, bacteria, fungi, or even synthetic reservoirs. Each of these options has different chemical properties and extraction protocols, conferring various advantages and disadvantages depending on the desired application [31]. For example, there are significant differences between fungal eumelanin and synthetic eumelanin, likely due to the fact that in vivo melanogenesis is vesicle-associated and enzymatically catalyzed, while synthetic melanogenesis often relies on spontaneous autopolymerization in solution [32]. In addition, synthetic melanins can be costly to produce [33]. Melanin from animal and plant sources can be less expensive to produce than the synthetic route, but it can be difficult to purify, as melanin is often associated with other biomolecules. Thus, microbial melanin is often touted as a low-cost and high-yield alternative to synthetic, animal, or plant melanin [33]. More specifically, fungal melanin is attractive due to the well-known protective functions in fungal organisms adapted to survive extreme environments [1].

Fungal organisms can produce different kinds of melanins (Figure 2). Most fungal melanins are allomelanins, which do not contain nitrogen [34]. DHN melanin is a common type of allomelanin derived from the polymerization of 1,8-dihydroxynaphthalene (DHN) [35]. The polyketide pathway that produces allomelanins begins with an endogenously produced molecule of acetyl coA or malonyl coA, which undergoes several reductions and dehydrations to produce 1,8-DHN [36]. The complete literature for scientific studies of fungal melanins dates back to the 1960s [37].

Some basidiomycetous fungi, such as *Cryptococcus neoformans*, produce eumelanin [38]. Eumelanins are derived from the amino acid tyrosine, and unlike allomelanins, they contain nitrogen [39]. Fungi synthesize eumelanins via the L-3,4-dihydroxyphenylalanine (L-dopa) pathway, which begins by using laccase to oxidize L-dopa into dopaquinone [40]. Eventually, the pathway produces dihydroxyindoles that can polymerize into eumelanins [40]. *Cryptococcus neoformans* cannot carry out this pathway without an exogenous substrate [40]. Pyomelanins are also derived from tyrosine, but they are produced by fungi such as *Aspergillus fumigatus* via the tyrosine degradation pathway, which involves the oxidative phosphorylation of homogentisate (HGA) [41].

Although not heavily discussed in recent reviews, some mushrooms and basidiomycetes have been shown to produce melanin from the precursor glutaminyl-hydroxy-benzene (GHB) [42]. The presence of GHB on mushroom caps of *Agaricus biosporus* has been correlated with their susceptibility to browning, and the transformed product is commonly dubbed GHB-melanin [43]. Although a synthesis pathway has not been fully outlined, it is thought to occur as GHB is transformed by a polyphenol oxidase (typically tyrosinase) into 2-hydroxy-*p*-iminobenzoquinone (2-HpIBQ) and polymerized [42,44]. The diversity of pathways that organisms use to synthesize melanin explains in part why these polymers are not as well understood as other biopolymers such as proteins or nucleic acids [45].

When extracting fungal melanins for human applications, eumelanins are often preferred as allomelanins are attached to the inner side of the fungal cell walls [46]. In contrast, eumelanins are used by fungi to neutralize toxic environmental compounds, making these melanins easier to extract extracellularly [46]. Currently, there are several different methods used to culture melanotic fungi, but most methods use a combination of tyrosine and metal ions [46]. Recently, fungal strains were genetically modified to become more melanotic for extraction purposes, principally by overexpressing genes for tyrosinases [33].

Prior to melanin extraction, isolating melanotic yeast can be a challenge depending on the yeast used. Some melanized yeasts are polyextremophiles, which can be difficult to isolate due to their slow growth rates and low competitive ability [47]. However, researchers have proposed the method of enriching black extremophile yeast of the order Chaetothyriales on aromatic hydrocarbon, which can inhibit the growth of other microbial species while allowing black fungi to be cultured [47].

Another challenge associated with the industrial use of fungal melanin is the potential pathogenicity of fungal species. This requires scientists to identify non-pathogenic species of melanized fungi, attenuate pathogenic melanotic fungi, or to introduce melanization to a non-pathogenic species [48]. Alternatively, when using pathogenic species for melanin extraction, operations must be carried out in laboratories certified for a given biosafety level.

While the challenges of fungal melanin extraction have made industrial-scale implementation difficult in the past, newer extraction techniques are simpler and produce higher yields of the compound [49]. For example, a 2015 paper recorded a 10% yield of fungal melanin from *Auricularia auricula* after several steps, including treatment with lytic enzymes, guanidinium thiocyanate, chloroform, and HCl [50]. In contrast, a 2019 paper recorded that an optimized strain of *Armillaria cepistipes* was able to produce a 99% yield of melanin following a simpler extraction procedure [49].

This review provides an overview of the various developments attempting to use specifically fungi-derived melanin for human application in fields including healthcare, bioremediation, and industry. The outlined projects are in varying stages of development, mostly due to the complicated nature of melanin extraction.

## 3. Health

Melanin’s unique properties allow it to have diverse applications in the field of human health, whether in a pharmaceutical, medical device, or antimicrobial. One of the most heavily studied health applications of melanin involves protection from radiation, given that fungal melanin is known for its radioabsorptive properties.

For example, in an experiment where mice were fed black mushrooms *Auricularia auricila-judae* and soon after irradiated, they tended to exhibit improved survival than the control over the course of 45 days [51]. Researchers postulated that melanin’s ability to dissipate Compton electron energy and scavenge free radicals would shield the mice’s gastrointestinal (GI) tract, preventing cellular apoptosis. When mouse GI tissue was examined 24 h post-irradiation, researchers found fewer apoptotic cells in the tissue of mice that had been fed black mushrooms. In addition, mice fed white mushrooms supplemented with melanin had the same outcome as those who were given melanized mushrooms to begin with, pointing to the role of melanin in the increased survival rates. However, because melanin is insoluble, its protective effects remained mostly limited to the gastrointestinal tract [51]. Melanin extract from *Auricularia auricula* also showed promise in reducing oxidative stress and enhancing survival in liver cells exposed to high doses of ethanol, providing a theoretical basis for the substance’s ability to treat alcoholic liver disease. The investigators associated their results with the liver cell’s activation of the antioxidase Nrf2 and the inhibition of the cytochrome CYP2E1, which produces ROS as it metabolizes ethanol. Additional investigations would further elucidate the pathway producing this phenomenon [52].

In a separate mice study, those given melanin from the fungus *Gliocephalotrichum simplex* not only experienced better survival from irradiation, but also showed improvement in spleen parameters, reduced oxidative stress in the liver, and reduced production of inflammatory cytokines [53]. The authors suggested that a key mechanism of melanin’s protective property in the study was its ability to reverse the decrease in phosphorylation of the transducing protein ERK that is commonly seen upon radiation exposure [53]. Radioprotective technologies are needed for the protection of multiple vulnerable demographics. For example, radiation can have a harmful effect on patients receiving it for diagnostic or therapeutic purposes [51]. Cardiac diagnostic procedures alone account for about one-fifth of the radiation exposure per person per year in the US [54]. In addition, some occupations receive high levels of radiation exposure, including healthcare professionals [54] and military personnel [55].

Currently, melanin has been proposed as a potential vehicle to protect astronauts from space radiation [56]. Particulate radiation is of particular concern in space travel, especially as it can generate secondary radiation upon interaction with spacesuits or spacecraft components [56]. The idea of fungal melanins as a potential material to protect against dangerous radiation in space originated from a series of studies demonstrating the ability of fungal melanin to attenuate and protect against different types of ionizing radiation [4,57,58,59]. A comparison of melanized and non-melanized forms of *Cryptococcus neoformans* and *Cryomyces antarcticus* found in both cases that the melanized cells were more resistant to a deuteron dose nearly 300,000 times higher than the dose lethal for humans [4]. In addition, samples of *C. antarcticus* were exposed for over a year on the International Space Station (ISS) to the radiation conditions in low Earth orbit. The samples not only survived, but also maintained a mutational load below 5% and sustained metabolic activity [60]. The organism’s thick layers of melanin have been implicated as a possible explanation for its survival [60]. Given melanin’s ability to protect microorganisms from both ionizing and particulate radiation, it may be a favorable material for protection in manned space travel [56]. For example, a recent preprint concluded that a melanotic *Cladosporium sphaerospermum* plate on the International Space Station produced attenuated radiation levels compared to a control plate [61]. However, more research about the radioprotective properties of melanin is needed to confirm that the application of melanin to space materials is possible and beneficial.

Allomelanin has recently been investigated for both its porosity and its ability to absorb nerve gas stimulants in solution [62]. Given fungal melanin’s natural ability to absorb harmful materials while allowing essential cellular materials such as water and nutrients to pass through, the material’s intrinsic porosity is a valid possibility. In addition, melanin’s extensive binding capacity can permit it to bind to many possibly harmful materials. Researchers worked with both synthetic allomelanin derived from 1,8-DHN and natural melanin ghosts derived from fungal cells. While the melanin ghosts provided a higher binding capacity for paraoxon and diazinon than some synthetic allomelanins, they were not as porous as the synthetic analogues [62]. In addition, investigators noted allomelanin’s potential in protective, breathable fabrics by applying synthetic melanin on nylon-cotton fabric. Researchers also noted that the obtained melanin ghosts contained other polysaccharides such as chitin and glucan, making it more difficult to examine the properties of natural allomelanin [62].

Although not specifically for radioprotection, the melanized medicinal mushroom *Inonotus obliquus* has shown in vitro effects against tumor growth and diabetes mellitus [63]. In one investigation, B16-F10 melanoma cells that were exposed to aqueous extracts of *I. obliquus* for 48 h exhibited decreasing, dose-dependent viability [64]. The study also noted that the antitumor properties performed in vivo as well; mice that received the extract through an intraperitoneal route showed inhibited tumor mass growth. These effects were associated with G0/G1 arrest through the down-regulation of p53, pRb, and p27 proteins, although the investigators acknowledged the need for further research to elucidate a complete mechanism of action [64]. Thus, it is possible that melanin may not be a contributor to the extract’s antitumor properties. However, melanin’s presence in *I. obliquus* may still have health benefits. Water-soluble melanin complexes from the organism have shown insulin-sensitizing activity [65]. When the melanin complex was extracted via filtration and rotary evaporation, it reduced adiposity in high-fat obese mice [65].

Fungal melanin has also been implicated to have potential use in the functional biointerfaces used for stem cell manipulation [66]. Eumelanins were proposed for this purpose due to their antioxidant and electrical properties. However, eumelanin can be vulnerable to degradation from alkaline or oxidative stress. Manini et al. propose the alternative use of fungal allomelanins, citing the material’s resistance to degradation and smoothness [66]. Such a mycomelanin film was able to encourage the differentiation of stem cells towards an endodermal lineage [66].

## 4. Bioremediation

Melanized fungi and fungal melanin have both been proposed as mechanisms for removing various toxins from a polluted environment. Fungi can degrade volatile organic compounds (VOCs), and the ability of melanotic fungi to survive acidic or dry conditions makes them great candidates for VOC absorption [67]. In western countries where individuals spend much of their time indoors, VOCs emitted from construction materials, appliances, and cleaning chemicals have been implicated as a cause of sick building syndrome (SBS) by the United States Environmental Protection Agency (EPA) [68]. Researchers found that melanized fungal cultures placed in an indoor air model were able to reduce the VOC content by more than 96% after 48 h [69]. However, researchers noted that which melanotic fungal species were used was an important factor for the feasibility of using fungi to eliminate VOCs in indoor spaces [69]. For example, some species produced putative volatile metabolites in low concentrations [69]. In addition, fungal species prone to rapid growth and/or sporulation may become prone to aerosolization, presenting challenges both as an allergen and as a pathogen to immunocompromised individuals [69]. The researchers ultimately concluded that slow-growing melanotic fungi may be optimal for future applications [69].

Melanin is also known to have properties that facilitate the absorption of heavy metals, making it a good candidate for addressing heavy metal pollution in waterways and the environment. One natural example of this is that of fungi isolated from a uranium mine in Brazil [70]. Some of these species showed changes in melanin production, and most of these species were deemed to have a high potential for uranium absorption from water upon a biosorption test [70]. In a different setting, nanofiber membranes made with fungal melanin from *Armillaria cepistipes* were able to absorb heavy metals from water [71]. With these particular membranes, the fungal melanin was better able to absorb heavy metals over essential metals [71]. Fungal melanin derived from *Amorphotheca resinae* for the purposes of heavy metal absorption was able to be reused, pointing to the economic advantages of fungal melanin for removing metals from the environment [72]. After metal ions were absorbed in a pH of 5, exposing the melanin to acidic conditions allowed the metal ions to dissociate. For up to five cycles of binding and eluting these metals, the melanin’s binding capacity was maintained [72].

Fungi can also play a major role in the absorption of nuclear pollution, given their resistance against and sometimes affinity for radiation. Fungal growth has been detected in and around the Chernobyl Nuclear Power Plant [73] and some fungi have been documented growing towards sources of radiation in a sort of radiotropism [74]. The extensive surface area of hyphae in saprotrophic fungi makes them excellent absorbers of radionuclides in the environment [75]. In one experiment, *Rhizopus arrhizua* and *Aspergillus niger* were able to remove 90–95% of radiothorium from solution [76]. One issue with using fungi for radioabsorption involves the concern that radioactive compounds could be transferred up the food chain from wild mushrooms to animals and people [77]. This was the case following the 1986 Chernobyl accident, in which cesium-137 remained highly concentrated in the fruiting bodies of edible fungi. In parts of Eastern Europe and even Great Britain, wild fungi consumption contributed to human intake of radioactive cesium [77]. However, this does not rule out the possibility of using extracted fungal melanin for radionuclide absorption applications.

## 5. Industry

Fungal melanin has found one potential application in the packaging of pork lard. Most commonly, fatty products such as lard go rancid due to the oxidation of lipids [78]. Luposiewicz et al. added fungal melanin to the gelatin coatings of pork lard and found that lard with the modified coatings tended to have lower oxidative rancidity [78]. This may be due to melanin’s antioxidant properties counteracting the effect of oxygen free radicals in the environment [78].

Polylactic acid is a promising bioplastic, but its use is currently limited in food packaging applications due to its low thermal stability and solvent resistance [79]. When modified with fungal melanin, polylactic acid demonstrated improved barrier properties [79]. However, these properties decreased when too much fungal melanin was added [79].

Given its distinctive black color, microbial melanin has clear potential in industrial dyes. The study of natural dyes is growing in popularity, as they have been seen as safer and more environmentally friendly than synthetic dyes for food, textiles, and materials [80]. Researchers were recently able to dye poplar veneer using melanin secreted by *Lasiodiplodia theobromae* [81]. Beyond aesthetic effects, this dyeing of wood may be able to mimic the color of more expensive tree species while utilizing more efficient, faster-growing trees [82]. While the experiment shows that such a method is feasible, more data are needed to determine the fastness of the dye, or the material’s ability to maintain its color over time [81].

Although the literature on natural fungal melanin’s use in textiles is limited, investigators recently examined the ability of synthetic allomelanin to adhere to nylon-cotton fabric swatches [62]. Melanin derived from other microbial species, such as *Streptomyces spp*., has successfully dyed wool without mordanting [83]. Barriers to natural fungal melanin’s widespread use in textiles may include the size of the polymer and the need for large-scale extraction mechanisms.

## 6. Conclusions

Melanin has played a function in stress responses in biological organisms long before humans took notice of it, and the scientific community has yet to discover many important properties of microbial melanin, including its structural characterization and large-scale extraction. The unique properties fungal melanin exhibits in nature can be applied to human use, and such applications are numerous and imminent. However, the scientific community must also focus on confronting the challenges associated with adopting fungal melanin by identifying or creating non-pathogenic melanotic species, refining methods of isolating cultures, and improving methods of melanin extraction. The potential for microbial melanin’s use to improve human health, environments, and industries further emphasizes the importance of further study into these pigments from multiple fields of interest.

## Figures and Tables

**Figure 1 jof-07-00488-f001:**
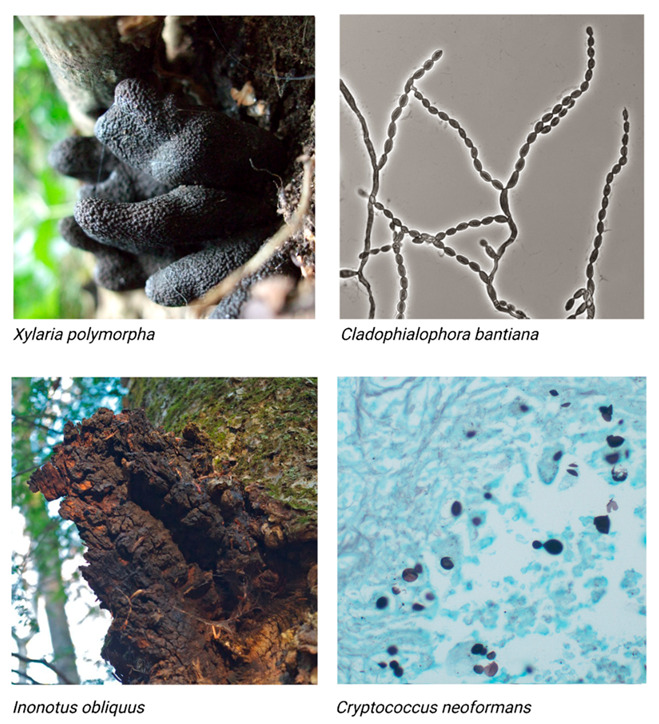
Panel of four images of different melanized fungi. *Xylaria polymorpha* (top left), a mushroom also referred to as ‘Dead Man’s Fingers.’ The stromata of *X. polymorpha* have an average total length, including rooting bases, of 5 to 8 cm by an average 2 cm diameter [27]. “Black fungi—*Xylaria polymorpha* (dead man’s fingers)” by ohi007 is licensed under CC BY-NC-SA 2.0; *Cladophialosphora bantiana* (top right), a pathogenic mold. *C. bantiana* conidia are approximately 5 to 10 μm in length [28]. “File:Cladophialophora bantiana UAMH10767.jpg” by Medmyco is licensed under CC BY-SA 4.0; *Inonotus obliquus* (bottom left), also known as the Chaga mushroom. Chaga appears as a sclerotia ranging from 5 to 40 cm in diameter [29]. “Chaga mushroom (*Inonotus obliquus*)” by Distant Hill Gardens is licensed under CC BY-NC-SA 2.0; *Cryptococcus neoformans* (bottom right), a pathogenic yeast. Cryptococcal cells range from 5 to 10 µm in diameter [30]. “Cryptococcosis—GMS stain” by Pulmonary Pathology is licensed under CC BY-SA 2.0. Images presented are not to scale.

**Figure 2 jof-07-00488-f002:**
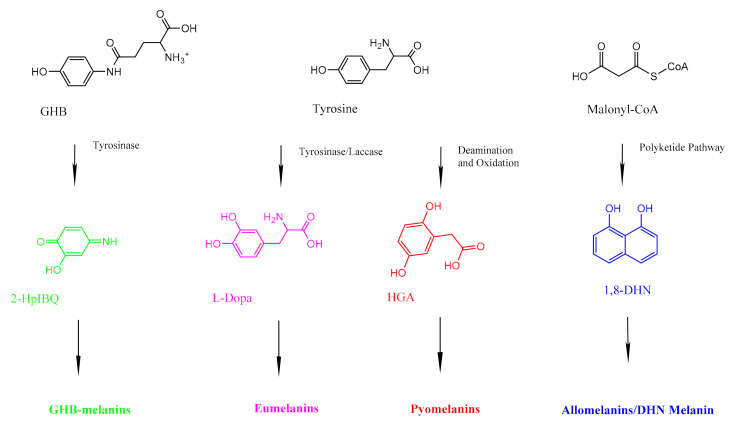
A simplified diagram showing the precursors for the three different kinds of fungal melanin. GHB-melanins (far left, green) are synthesized through a series of reactions with tyrosinase. Eumelanins (center left, pink) are synthesized via the L-Dopa pathway, which uses tyrosine as a precursor. Pyomelanins (center right, red) also use tyrosine as a precursor but are ultimately derived from HGA. Allomelanins (far right, blue) are derived from 1,8-DHN.

**Table 1 jof-07-00488-t001:** Examples of the functions of fungal melanin in living fungal organisms.

Function	Source
Photoprotection	[2,3,4]
Thermoregulation	[5,6,7,8]
Energy Harvesting	[5,22,23,24]
Metal Binding	[17,18,19]
Chemical Stress Response	[10,11,12,25]
Antioxidant	[7,9,26]
Anti-Desiccation	[14,26]
Virulence	[20,21]

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
