# Peer review of "Fungal Melanins and Applications in Healthcare, Bioremediation and Industry"

_jof, 2021, doi:10.3390/jof7060488_

Round 1

Reviewer 1 Report

Thank you for submitting an interesting and helpful manuscript on Fungal Melanins and Applications in Healthcare, Bioremediation and Industry. The analysis is well done, and some practical uses of melanotic fungi are listed very well. I strongly believe this topic is of interest to the readers during this late pandemic. Before making it published, I suggest the authors to address few typological errors and minor changes mentioned in the following

line 27: has to keep a full stop after [1]. 

line 36: provides

line 39: accompanies at

line 42: the citations [10] were repeated twice

line 80: typo should be"dihydroxyindoles"

line 93-100, under figure 2, all the scientific names of fungi should be italicized. Also, description for each one of the images should be separated by semicolon or it should come in the next line for easy understanding 

line 177: it should be proposed

line 240: Lasiodiplodia theobromae should be italicized

line 249: spp., should be in italics . And typo on died, it should be "dyed"

Comments on citations: 

Few references lack full information in the citations section. For example, ref 65 doesn't have the details of which year, volume, pages. 

Ref 69, 28 lacks the same. 

Check all the citations again to keep consistency. 

I think you are on the right track to provide this manuscript to the readers at the right time. I congratulate the authors for their work. 

Author Response

Reviewer 1:

Thank you for submitting an interesting and helpful manuscript on Fungal Melanins and Applications in Healthcare, Bioremediation and Industry. The analysis is well done, and some practical uses of melanotic fungi are listed very well. I strongly believe this topic is of interest to the readers during this late pandemic. Before making it published, I suggest the authors to address few typological errors and minor changes mentioned in the following

line 27: has to keep a full stop after [1]. 

Response: Accept

Action taken on revision: Changed accordingly.

line 36: provides

Response: Accept

Action taken on revision: Changed accordingly

line 39: accompanies at

Response: Accept

Action taken on revision: Changed accordingly

line 42: the citations [10] were repeated twice

Response: Accept

Action taken on revision: Changed accordingly

line 80: typo should be "dihydroxyindoles"

Response: Accept

Action taken on revision: Changed accordingly

line 93-100, under figure 2, all the scientific names of fungi should be italicized. Also, description for each one of the images should be separated by semicolon or it should come in the next line for easy understanding 

Response: Accept

Action taken on revision: Changed accordingly

line 177: it should be proposed

Response: Accept

Action taken on revision: Changed accordingly

line 240: Lasiodiplodia theobromae should be italicized

Response: Accept

Action taken on revision: Changed accordingly

line 249: spp., should be in italics . And typo on died, it should be "dyed"

Response: Accept

Action taken on revision: Changed accordingly

Comments on citations: 

Few references lack full information in the citations section. For example, ref 65 doesn't have the details of which year, volume, pages. 

Ref 69, 28 lacks the same. 

Check all the citations again to keep consistency. 

Response: Accept

Action taken on revision: Changed accordingly

I think you are on the right track to provide this manuscript to the readers at the right time. I congratulate the authors for their work. 

Response: The authors would like to extend their genuine gratitude for the kind and constructive comments from this reviewer.

Reviewer 2 Report

A broad but well rounded review of fungal melanins. The literature hit all major areas and had some interesting sub-discussions. The writing is solid and engaging. I see no major issues.

  • line 28: ultraviolet shouldn't be capitalized

Reviewer 3 Report

General comments

This is a new review on melanin written by a group with expertise in fungal melanin. I think the review is interesting and useful for melanin researchers, as it covers some aspects that had not been discussed in previous similar reviews and focused to 3 important fields (Health, Bioremediation and Industry). The review indicates some peculiar properties of melanin that are not in other reviews.  Among these peculiar aspects (potential vehicle to protect astronauts from space radiation or absorber of nerve gas, some of them are difficult to believe, and they should be discussed or assessed in addition to the mere citation. For instance, it is hard to believe the conclusion of ref. 43 (lines 137-138, mice fed with black mushrooms Auricularia auricila-judae tended to survive the effects of irradiation better than the control. I have some doubts about the conclusion cited as ref. 51 by own experiments (line 170). Authors should include some comments about these conclusions. At least, I suggest mentioning that further experiments are needed about that extraordinary melanin effects.

On the other hand, it is good to see that the methods to extract fungal melanin are improved, as mentioned at lines 125-126. However, bacterial melanins are obtained also with high efficiency, and synthetic melanin can be easily synthesized. Some comments about the comparison and advantages of fungal melanins versus bacterial or synthetic melanin would be helpful.

Particular comment

Concerning Section 2, Fungal Melanins and Figure 1

Some fungi, such as the edible Agaricus bisporus, are able to form a structurally different type of melanin that could be considered related to dopa-melanin (or allomelanin depending of the used criteria) but they are not usually considered in previous reviews. This is also the case of the current manuscript. They are related to Gamma-glutaminyl-4-hydroxybenzene (GHB). You can find information about that in some original papers (i.e. http://dx.doi.org/10.1016/j.fgb.2012.10.004) or melanin reviews (i.e. http://dx.doi.org/10.1155/2014/498276). As this type of melanin is missing in this review, Figure 1 could be considered not completed. I think that it would be an improvement to include it, as this is a new review focused to fungal melanins. Agaricus bisporus is interesting for health rahter than bioremediation or industry. 

Minor points

Line 80: replace dihydoxyindoles by dihydroxyindoles

Please, complete reference 28

Round 2

Reviewer 3 Report

Excellent reply to reviewers and brilliant and informative modifications. As the authors write in the reply cover letter, I also think that the paper has been improved. Some points have been clarified and completed.